# Ecological niche modeling as an effective tool to predict the distribution of freshwater organisms: The case of the Sabaleta *Brycon henni* (Eigenmann, 1913)

Daniel Valencia-Rodríguez[1]*, Luz Jiménez-Segura[1], Carlos A. Rogéliz[2], Juan L. Parra[3]

**1** Departamento de Biología, Grupo de ictiología de la Universidad de Antioquia, Universidad de Antioquia, Medellín, Antioquia, Colombia, **2** The Nature Conservancy, Bogotá, Cundinamarca, Colombia, **3** Departamento de Biología, Grupo de ecología y evolución de vertebrados, Universidad de Antioquia, Medellín, Antioquia, Colombia

\* davarod@gmail.com

## Abstract

Ecological niche models (ENMs) aim to recreate the relationships between species and the environments where they occur and allow us to identify unexplored areas in geography where these species might be present. These models have been successfully used in terrestrial organisms but their application in aquatic organisms is still scarce. Recent advances in the availability of species occurrences and environmental information particular to aquatic systems allow the evaluation of these models. This study aims to characterize the niche of the Sabaleta *Brycon henni* Eigenmann 1913, an endemic fish of the Colombian Andes, using ENMs to predict its geographical distribution across the Magdalena Basin. For this purpose, we used a set of environmental variables specific to freshwater systems in addition to the customary bioclimatic variables, and species' occurrence data to model its potential distribution using the Maximum Entropy algorithm (MaxEnt). We evaluate the relative importance between these two sets of variables, the model's performance, and its geographic overlap with the IUCN map. Both on-site (annual precipitation, minimum temperature of coldest month) and upstream variables (open waters, average minimum temperature of the coldest month and average precipitation seasonality) were included in the models with the highest predictive accuracy. With an area under the curve of 90%, 99% of the species occurrences and 68% of absences correctly predicted, our results support the good performance of ENMs to predict the potential distribution of the Sabaleta and the utility of this tool in conservation and decision-making at the national level.

## Introduction

Freshwater habitats are key areas for biodiversity [1]. They harbor 9.5% of all described species, but only occupy 0.01% of the Earth's surface [2]. Many streams and lakes are considered biodiversity hotspots and understanding the biogeography of freshwater organisms is key for the

**Funding:** The research of DVR, LJS, CAR and JLP was supported by the Empresas Publicas de Medellin (www.epm.com.co) and Universidad de Antioquia (www.udea.edu.co) in the framework of the agreement (CT-2017-001714). DVR receives financial support from this agreement for the payment of registration and maintenance fees. The funders had no role in study design, data collection and analysis, decision to publish, or preparation of the manuscript.

**Competing interests:** The authors have declared that no competing interests exist.

conservation and management of biodiversity. Geographic distributions of freshwater organisms are necessary inclusions to achieve these goals. Although there has been recent increased interest in modelling the geographic distribution of freshwater species [3–5], the majority of studies that use geographic distributions to establish macroecological patterns [6] or assess the conservation status of species [7], are based on the identification of basins where there is evidence of occurrence.

Representing the distribution of freshwater species as occupied basins has many logistical advantages and often just requires a good elevation layer to map the drainage network and identify those basins where the species of interest is present. Nevertheless, the scope of analyses using this strategy has limitations in freshwater systems: the whole basin is not inhabited by fish and the area actually occupied is much less than the basin's total area (S1 Fig). Additionally, alluvial channel patterns (straight, meandering, braided, island-braided and anastomosing) [8] and drainage network density are variable within and between basins. This diversity in geomorphic configurations creates a changing mosaic of habitat patches with different ages within the drainage network [9], limiting the conclusions obtained from the mapped distributions of species (S1 Fig). Generating representations well-suited to the areas actually available and occupied by fishes is an achievable goal nowadays, with favorable implications for conservation and research [10].

Ecological niche models (ENMs) have been successfully used in terrestrial organisms in order to obtain potential distributions, but their application to aquatic organisms is scarce. Nowadays, the availability of occurrence records and specific variables for aquatic systems allow the application and evaluation of these models [11, 12]. Ecological niche modelling in freshwater organisms is a challenge for four reasons: their movements are restricted to the drainage network [13], species occurrence is influenced by the physicochemical characteristics of the water [14], physiographic variations within the channel can occur in very short stretches [13], and the conditions of the location where the individual is reported is influenced by the upstream conditions of the tributary. Many of these gaps have begun to be filled at the global level with the generation of high-resolution environmental layers at both the spatial and temporal scale. For example, at a global resolution of ~1 km$^2$ there are raster layers with information regarding climate, topography and surrounding vegetation cover [3] and, recently, global maps of stream flow have been generated [12]. These layers have rarely been used in neotropical systems; therefore, their use can help to improve the validation of current hypotheses regarding the distribution of freshwater organisms.

As a model species to assess the effectiveness of ENMs in Andean freshwater systems, we used the Sabaleta *Brycon henni* Eigenmann 1913, an endemic fish of the Colombian Andes occurring between 300 and 2000 meters of elevation within the Magdalena River Basin (Fig 1). This species is considered of minor concern (LC) by the International Union for Conservation of Nature (IUCN) [15]. The Sabaleta is an omnivorous fish [16] showing short movements between river channels and the streams that flow to them, associated with precipitation seasonality [17, 18]. Additionally, the quality of its meat makes it an attractive protein source for the indigenous communities and the rural populations, particularly those located in mountainous regions. Thus, knowing the distribution of this species is key for the development of adequate management strategies for future studies in conservation, protection, recovery and use of freshwater fish species.

This study characterizes the niche of *B. henni*, using ENMs that incorporate variables specific to freshwater systems to predict its geographic distribution across the Magdalena River Basin hydrological network. The results are then compared to the current distribution hypothesis of the IUCN. Our goal is to promote the use of ENMs for the study of freshwater organisms in neotropical systems.

## Methods

### Ethics statement

This study was carried out with recommendations and approval of the Ethics Committee for Animal Experimentation from the Universidad de Antioquia (CEEA). Protocol was reviewed

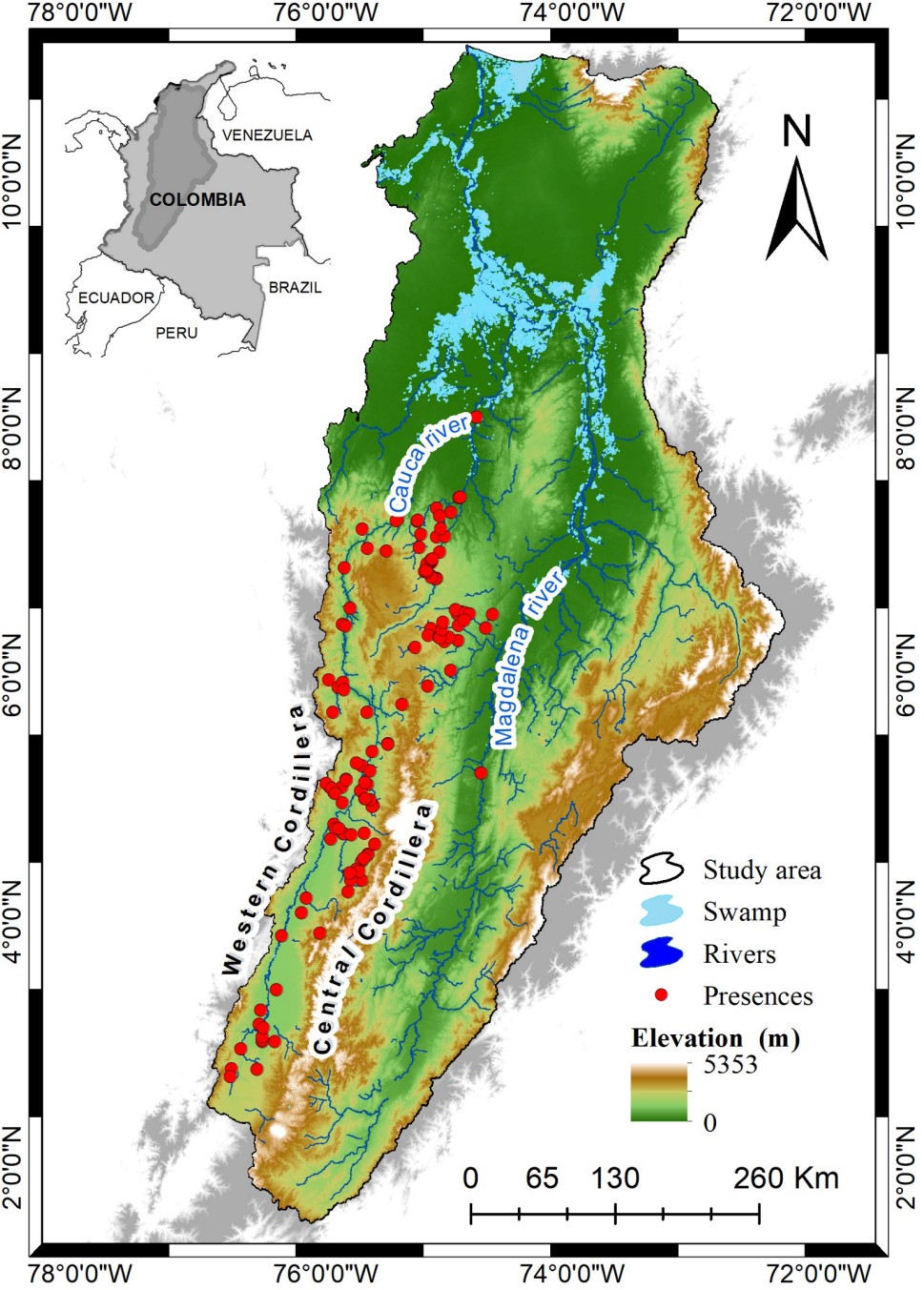

**Fig 1. Geographic location of the study area.** Blue represents the main bodies of water made up by rivers and swamps, red dots represent occurrence records and color scale elevation ramp shows topographic features of the Magdalena basin. The digital elevation model (SRTM, 1 arc-second) was obtained from USGS Earth Explorer (https://earthexplorer.usgs.gov) and the rivers, basins areas and swamp shapefile from IGAC (https://geoportal.igac.gov.co). All other products were produced by the authors and are copyright-free.

and approved in November 14 of 2017 by CEEA and the investigation was approved in December 7 of 2017.

## Accessible area

The accessible area is defined as all those areas where an organism of interest, in this case *B. henni*, could have access through its means of dispersion within a wide temporal interval [19]. The Magdalena River Basin involves two of the largest rivers in Colombia (Magdalena and Cauca Rivers), it has a bimodal precipitation regime, important elevation gradients generating a high diversity of landscapes and climatic conditions where most of the hydroelectric power generation has been concentrated in the last 40 years [20]. The species occurrence records within the basin are mainly found along the Cauca River, and a few of them in the Magdalena River (Fig 1).

## Occurrence records

Occurrence records of *B. henni* were obtained from the icthyology collection database of the Universidad de Antioquia (CIUA), along with cured historical records found in the Global Biodiversity Information Facility [21]. Additionally, field visits were made to different locations along the Magdalena Basin from July 2018 to July 2019 (S1 Table). All visited locations were georeferenced with a GPS Garmin 64s. Records were mapped using ArcMap 10.2 [22] and its validity verified according to the taxonomic and geographic information found in the list of freshwater fish species of Colombia [23]. Records that did not match the bodies of water were transferred to the nearest cell with a maximum radius of 500 m; whenever a record exceeded this threshold, it was discarded. Each record was reviewed to check the correspondence of locality description (e.g. department, municipality and elevation) with the georeference.

Only records obtained since 1950 were used, since this is the first year from which the climatic variables have information [24]. The final database contained 186 records after discarding duplicate records (all records within the same 30 second resolution pixel), dubious records from outside the Magdalena River Basin or whose description (e.g., municipality or department) did not coincide with the assigned coordinates (435 records discarded in total).

## Environmental variables

A total of 87 environmental variables were originally considered at 0.083 arc spatial resolution (~1 km$^2$ near the equator). These included layers of topography (slope [˚C]*100), climate (e.g., temperature [˚C]*10 and precipitation (mm)–on site and aggregated upstream), and land use (percentage of broadleaf trees, percentage of population centers and percentage of open waters) from the EarthEnv project [3]. Variables with local information obtained from the WorldClim dataset [24] (e.g., mean annual temperature at one particular cell) and upstream of the basin (e.g., average of mean annual temperature across all cells upstream a particular cell of interest) were of special interest since these variables have only been proposed as useful for organisms that inhabit this type of ecosystems, where the conditions in a locality are dependent on the characteristics upstream from it [25]. Therefore, one particular interest of this work is to evaluate the relative contribution of these variables to the model. River flow data comes from the multiannual averages of the maximum, minimum and average flow [12]. This variable is of particular importance because freshwater species are confined to bodies of water with singular features, and river flow is a direct conditioner for these characteristics, temperature, pH, dissolved oxygen, substrate and surrounding vegetation, for example [26].

Due to the high number of environmental variables available for each category (topographic, land use, climatic data points and upstream basin), an initial selection was made

based on the intensity of their correlation (r > 0.75). Within each category, groups of correlated variables were identified, and we selected one based on its biological significance according to the available literature [27–29]. We then grouped the set of selected variables and performed an additional correlation analysis to determine if there was a correlation among all variables (S2 Fig), including those that represented the value in each cell and those that represented the value upstream of the basin, resulting in a total of 23 environmental variables (See S2 Table).

## Ecological niche model

With the occurrence records obtained from the information repositories and the mentioned environmental variables, we modeled the potential hypothetical distribution of *B. henni* with the Maximum Entropy algorithm using the software MaxEnt version 3.4.1 [30]. This algorithm was chosen because it only requires occurrence records and its application in previous studies has given good results [30, 31]. MaxEnt uses occurrence records together with a characterization of the available environments in the accessible area (background) to identify which environmental conditions are favored by the organism of interest. The background environment affects the model results and therefore is important to determine the accessible area and to recognize possible spatial sampling biases (e.g. rivers with a greater sampling effort). To overcome sampling biases, we generated a sampling effort layer in raster format with the same resolution of the environmental layers, which informs the algorithm of the relative sampling effort in each pixel. As a proxy for the effort we used the number of freshwater fish records present in GBIF for each cell. A value of 0.00001 was assigned to those cells with no records, so that the model could use them to characterize the environment, although with much less probability than those where some sampling was recorded. We used the ENMeval package to identify the regularization parameter value and the type of relationships to model, which minimized the omission rate and presented the best fit according to the Akaike information criterion [32]. We evaluated values of the regularization parameter ranging from 1 to 4 and lineal, quadratic, product, threshold, and hinge relationships. We validated each model with 20% of the occurrence records. MaxEnt's cloglog output was transformed to binary maps using the minimum training presence (MTP) threshold value since we are confident on the quality of the training data used. To obtain the potential distribution, we used ArcMap 10.2 [22] to reclassify the model of the species using as threshold the minimum probability value of suitable conditions present in the training records. The final result was a binary map with *B. henni* potential distribution (1–0; presence-absence).

## Relative importance of environmental predictors

To assess which environmental variables were the most informative for the model, we took two approaches: (i) we used the two methods available in MaxEnt to assess variable importance (percent contribution and permutation importance), and (ii) we generated a model using only on-site variables and compare its performance with the best model using both on-site and upstream variables. If both on-site and upstream variables were important for the model, we expect both types of variables to appear among the most informative predictors, and we expect a model including both types of variables to outperform all models using only on-site variables. The percent contribution method quantifies the contribution of each variable to the regularized gain of the model (i.e., the sum of each variable contribution must add to 100). The permutation importance method measures the drop in training AUC (in percentage) when permuting the values of each variable at a time with the background values. To obtain an estimate of the niche and its geographic prediction using only on-site predictors, we used the

same methodology as described above but only restricted to on-site variables. Through these two types of evaluations, we identified the most important variables within the ones evaluated in this exercise. We acknowledge that this exercise can identify the variables that lead to models with the highest predictive accuracy, but not necessarily identify the actual environmental tolerances of the organisms [33].

## Model evaluation

We used the additional occurrence records obtained in the field together with the absence records that we identified based on places with a large sampling effort where the presence of *B. henni* was not documented (a minimum of 6 years). This data set was used together with the model results to build a confusion matrix where the number of correctly predicted occurrences were quantified as true positives (a), the number of absence records that were predicted as occurrences were quantified as false positives (b), the number of occurrence records predicted as absences were quantified as false negatives (c), and the cells correctly predicted as absences were quantified as true negatives (d). From the confusion matrix we generated the ROC curve [34] that represents the sensitivity (proportion of correctly predicted occurrences, $a / [a + c]$) as a function of 1-specificity (proportion of absences predicted correctly) and we quantified the area under the curve (AUC), which is a measure of the model's performance in relation to a null model. Furthermore, we quantified Kappa (see S3 Fig) defined as the precision of the prediction in relation to a null model where points are randomly distributed [35], $[(a + d) — (((a + c) (a + b) + (b + d) (c + d)) / N)] / [N — (((a + c) (a + b) + (b + d) (c + d)) / N)]$, where N is the sum of all cases [36].

## Comparison with other models

To contrast the geographic prediction of our model with other existing species distribution hypotheses, we compared the binary map from the niche model with the range of *B. henni* published by the IUCN [37]. The mapping protocol used by the IUCN consists of collecting point locality data for each species, identifying all sub-basins at resolution 8 following the HydroBASINS database [38], and optionally, other sub-basins can be included where the species native presence is potential based on literature or expert knowledge [39]. We compared how much area was shared by both proposals and contrasted if the majority of records from *B. henni* from virtual repositories and field expeditions were found within both distribution proposals. To our knowledge, there are no other published proposed distributions for this species, except ones that indicate the whole country of Colombia [40].

## Results

We obtained a total of 607 records, from which we discarded 321 that were duplicates and 114 that we considered inaccurate and were not congruent with the current knowledge of the distribution (e.g., georeferences outside the Colombian Andes or in marine systems). Thus, we finally obtained a total of 186 occurrence records for *B. henni* (Fig 1). Most of the records were located in the western slope of the Central Cordillera and the eastern slope of the Western Cordillera. The detailed coordinates for each of these records are found in the S1 Table.

The best model according to AIC included lineal and quadratic features and a regularization parameter of 1.5. The model had a high AUC value (0.90; Fig 2B). The ROC diagram identified the point on the curve at which the convergence of sensitivity and specificity were maximized (0.82). The minimum value associated with a training occurrence record was 0.23; this value was used as a threshold for the binary map. Using this cutoff value, true positive rate

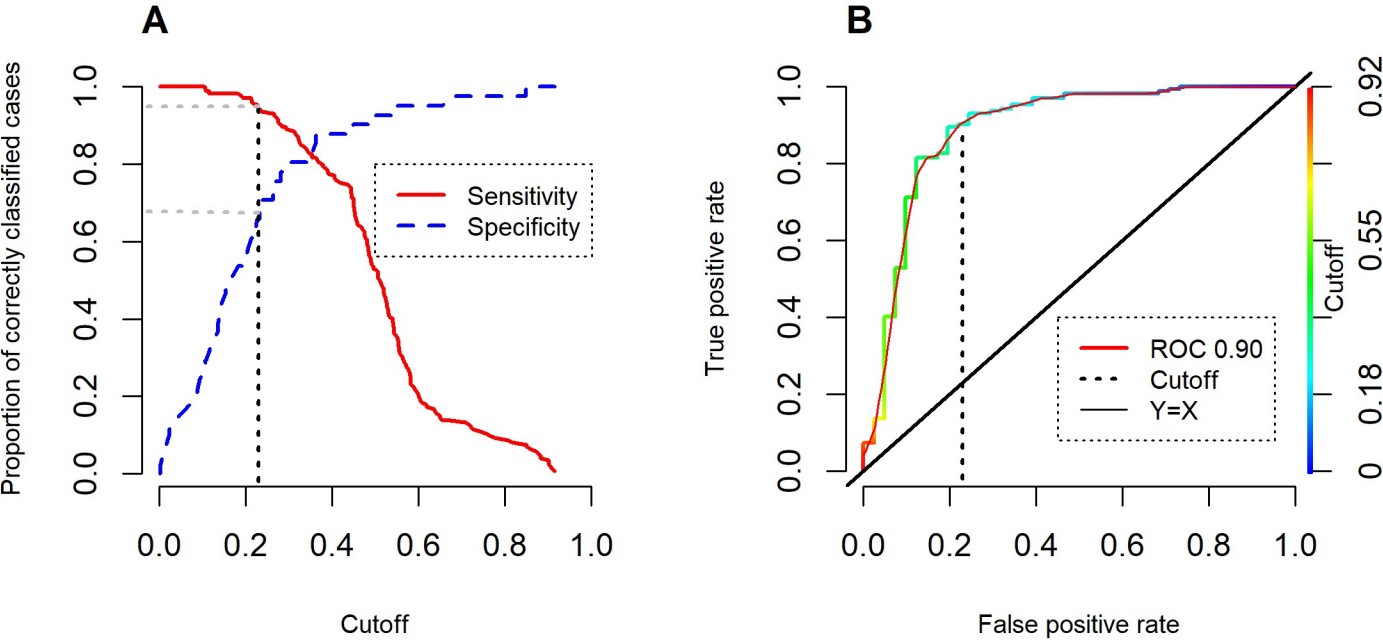

**Fig 2. Results of the model's performance.** (A) Sensitivity (true positive) and specificity (true negative) as a function of the cutoff value. (B) ROC curve generated from the distribution model of *B. henni* using an independent set of presence (156) and absence (42) points. Dotted vertical black line represents the cutoff value used to make the continuous model binary.

was greater than based on the convergence point (0.95 vs 0.82, respectively) and the true negative rate was lower (0.68 vs 0.82 respectively; Fig 2A).

By superimposing the binary and continuous output of the model we obtained the potential distribution for *B. henni* throughout the Magdalena Basin (Fig 3). We highlight the connection suggested by the model among bodies of water draining into the Cauca River, presenting a continuum of suitable environmental conditions for the species along the western slope of the central cordillera and the eastern slope of the western cordillera. The predicted distribution also includes tributaries of the Magdalena River, especially on the eastern flanks of the Central Cordillera.

Both on-site and upstream variables had a strong influence in the final model of *B. henni* (Table 1). Using the first approach by both methods to identify the relative contribution of predictors (percent contribution and variable importance), annual precipitation (bio_12) was identified as the most important variable, followed by the open waters' variable (Biolc_12) which represents floodplains in the Magdalena Basin. According to the best model, the Sabaleta prefers sites with higher annual precipitation (>2000 mm annual precipitation) and avoids mid to large floodplains (S4 Fig). Likewise, we obtained an important contribution of the minimum temperature of the coldest month (on-site, bio_6) and upstream basin (havg_6), followed by precipitation seasonality upstream basin (havg_15). The Sabaleta prefers sites with intermediate minimum temperatures between ~10 and ~23˚C and relatively colder sites upstream (notice a higher amplitude of the response curve mostly towards lower temperatures (S4 Fig). When restricting the model to only on-site variables, no model outperformed the best model generated with both on-site and upstream variables based on AICc nor AUC. The best

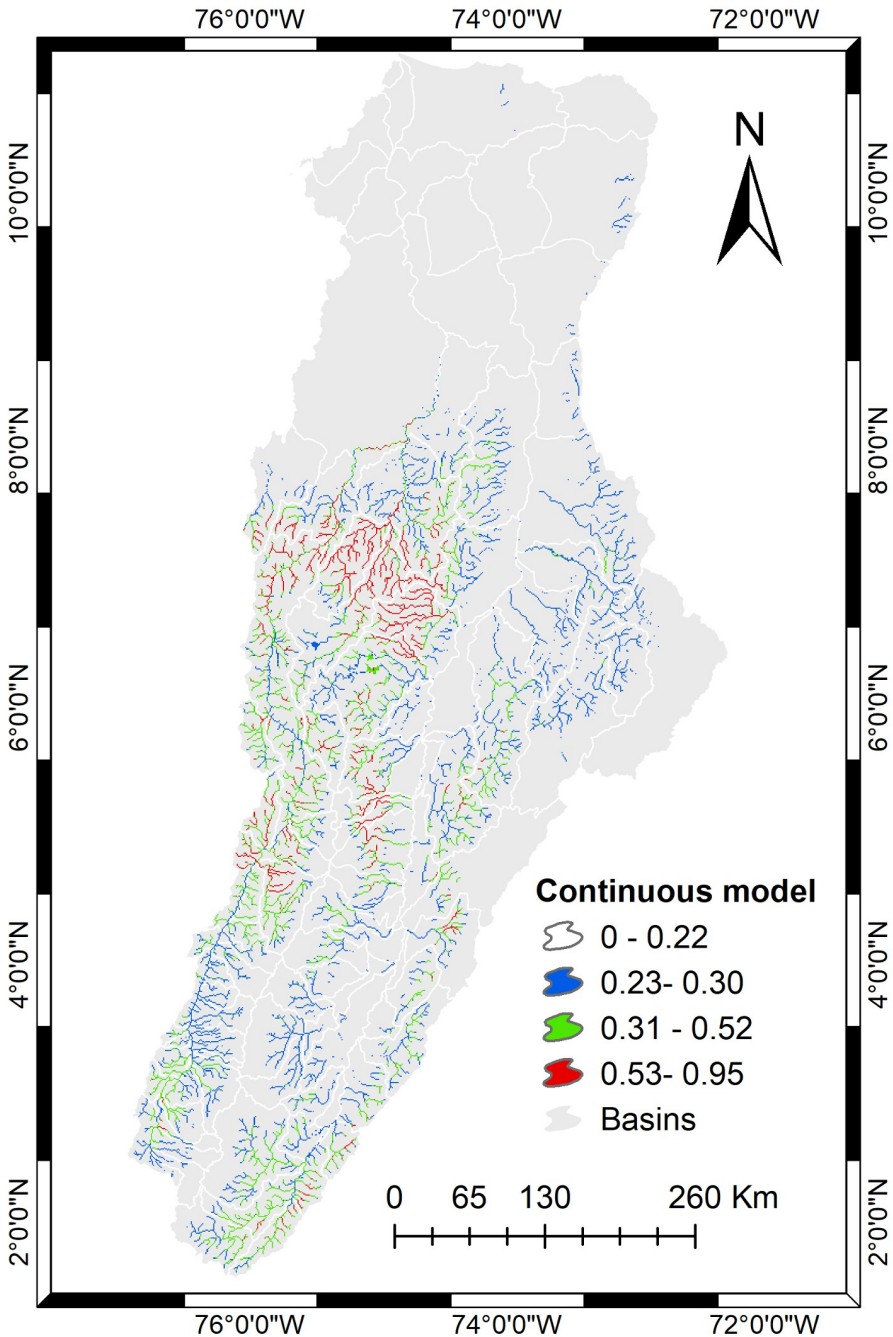

**Fig 3. Geographic projection of the continuous niche model (cloglog format) masked by the binary model to represent the potential distribution of *B. henni*.** White lines delimit basins and all colored watershed represent places with suitable environmental conditions for the species. The shapefile of basins was obtained from IGAC (https://geoportal.igac.gov.co/). All other products were produced by the authors and are copyright-free.

model using only on-site variables included more complex features than the best model using all variables (S3 Table). Response curves of the second-best model using only on-site variables were very similar to those estimated in the best model using both on-site and upstream variables (S4 Fig).

**Table 1. Percentage contribution (PC) and permutation importance (PI) of the most informative variables in the model.**

| Variable name | Importance | Criterion |
|---|---|---|
| Annual Precipitation | 28.5/18.4 | PC/PI |
| **Open Water** | 21.6 | PC |
| Minimum temperature of the coldest month | 23.6 | PI |
| **Minimum temperature of the coldest month upstream basin** | 20.5 | PI |
| **Precipitation seasonality upstream basin** | 11.8 | PI |

Both on-site and upstream basin (bold) variables were included in the best model within the five most relevant predictors according to either or both criteria.

The IUCN map covered approximately 25074 km$^2$ while our binary model of the potential distribution covered an area of 14646 km$^2$. By overlapping both proposals we obtained 6.8% overlap (Fig 4). However, when we verified the number of occurrence records used to evaluate the model (total of 30 occurrences) that were correctly predicted by each proposal, we obtained that the IUCN map was correct in 39% while our model was correct in 99% of the cases.

## Discussion

Our results support the successful performance of ENMs to predict the potential distribution of the Sabaleta and the potential utility of this tool for freshwater species in the field of conservation and decision-making at the national level. ENMs for freshwater species have improved their quality and utility with the generation of climatic and hydrological variables directly associated to watercourses [4]. Species distribution maps are essential for studies in biogeography and decision-making at the local and regional level [41, 42]. The result of the present study is a hypothesis for the potential distribution of *B. henni* from an ENM based on climatic and hydrological variables. This hypothesis must be further tested with occurrence and absence records and complemented with variables that describe the conditions of the river channels, physiological and connectivity studies, and surveys aimed to identify ecologically viable populations [43]. Nonetheless, we believe this is a robust evidence-based hypothesis that should be used for practical purposes.

The algorithm implemented in this study (MaxEnt) has been widely used in species distribution studies with a reasonable average discrimination [5, 10, 44, 45]. However, it is important to continue evaluating other algorithms that may outperform or complement MaxEnt [5]. The implementation of these models in conservation and decision-making can have important repercussions [41, 46]. In the case of *B. henni*, its distribution area would be 90% different from the one currently used to make conservation decisions (e.g. IUCN maps). In the following paragraphs we discuss the results found in relation to the model and its consequences for decision-making in conservation.

ENMs have been successfully used in biogeography and conservation [47, 48]. One advantage of these models is that they only require species occurrences and environmental variables, usually defined for large extensions and rough resolutions. This allowed us to identify which variables are informative for the model and how they behaved. In the case of *B. henni*, we evaluated the relative importance of particular variables (e.g. average annual temperature on a site) and variables summarizing the upstream conditions at a specific site (e.g. average temperature across all upstream locations). In the case of migratory fish, the latter variables should be important since they define the heterogeneity in a basin's section. These fishes use stretches containing a high heterogeneity since their reproductive cycles are associated to changes in the physio-chemical characteristics of water [49, 50]. Our results indicate that combining these

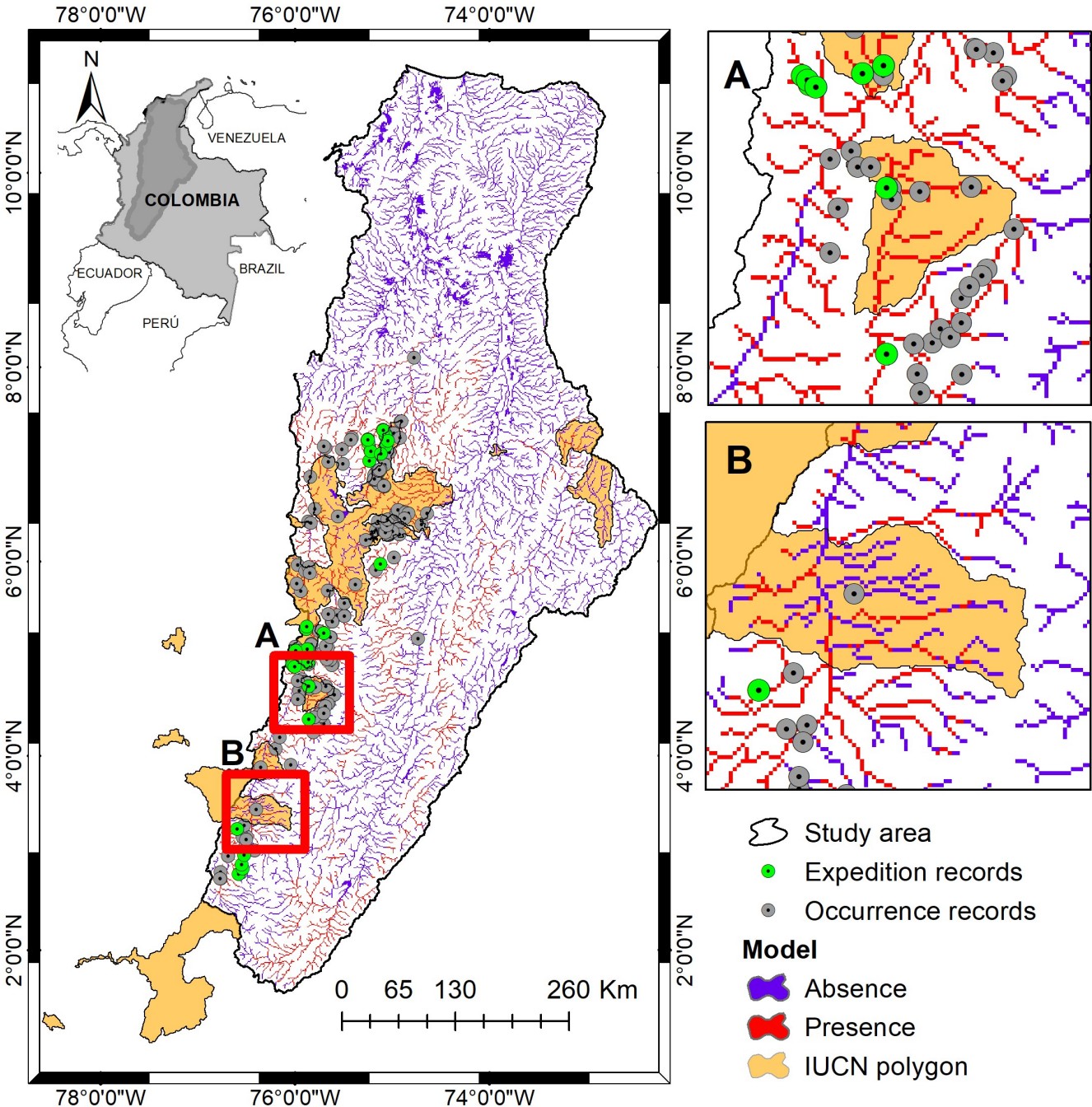

**Fig 4. Two alternative geographic distributions according to different methodologies.** The orange polygons represent the current distribution proposal according to the IUCN [37], in red the predicted areas as environmentally suitable for the species according to our niche model, the occurrence records are represented by gray points and green points are the records obtained during recent field expeditions. (A) Some areas predicted by the model but not included in the IUCN system were corroborated in the field with presence records of the species. (B) Some areas included in the IUCN system that were corroborated with presence data were not predicted by the model as suitable. The data used for this figure under CC BY license is granted permission from the International Union for the Conservation of Nature (IUCN), original copyright 2019.

environmental predictors (on-site and upstream conditions accumulated for the entire basin) is complementary and approximate better the distribution of these organisms. Despite the high correlation between on-site variables and the ones characterizing the upstream variation

(S2 Fig), we identified that both environmental predictors generate more accurate models in terms of AICc and AUC. Therefore, we suggest combining both types of data in niche modelling exercises.

The way in which occurrence probability is related to variables such as annual precipitation, open waters (e.g. floodplains), minimum temperature in the coldest month and precipitation seasonality (S4 Fig) agrees with previously reported physiological requirements of freshwater fish [18, 51, 52]. Response curves of these variables indicate that the Sabaleta can be found in sites with relatively high annual precipitation (> 2500 mm), little percent open water (<10%), profiles of minimum temperatures that include both low and high elevation sites (~10–20 C), and relatively little precipitation seasonality (S4 Fig). The importance of environmental variables associated to climate that turned out to be relevant for the distribution of *B. henni*, suggests that the species' life history is strongly associated with unidirectional Andean flow systems and a pronounced precipitation seasonality. *B. henni* is considered to be a short-distance migratory fish that, depending on the intensity of the rains, moves between creeks and river channels [17, 18]. Species with periodic strategies [53] are characterized by late maturation, high fecundity, low progeny survival and tend to inhabit highly seasonal environments so that individuals must move between different sites looking for favorable environmental conditions. These massive movements between environments are synchronized with climatic seasonality and are performed by part or by the entire population to carry out their reproduction [54]. In Andean aquatic systems, the hydrological pattern and the transport of nutrients generated by the rainy season are key to the selection of the optimal moment for reproduction [50, 55]. Most migratory fish species breed during the rains, some during the dry seasons, and others throughout the year. This variation is mainly influenced by the interaction between rainfall and specific environmental conditions in each type of aquatic system that offers food for adults and favors spawning success [53, 56]. Thus, while in the floodplains and the main channel of the river the favorable conditions for reproduction and recruitment are much more favorable during floods, in the Andean creeks, reproduction is much more common during dry season to avoid drift of eggs and larvae during rainy season [57], thus increasing embryo and larvae survival. For example, some characids like *B. henni* build nests to keep them safe from predators and the influence of currents. When embryos hatch, larvae develop a cephalic sucker to rocky substrata or stay in the pools to avoid flow [57].

In our case, we did not consider the anthropic influence in the Magdalena basin to obtain the distribution of *B. henni*. For example, we included occurrence records taken during the past decades and the climatic layers used represent conditions from 1950 to 2000 [24]. However, we included variables representing the predominant vegetation along the basins, but these were not especially informative. It would be possible to post-process the result of this model to crop out segments that are unsuitable due to anthropic influence [58]. It is worth noting that presence and absence data were used for the evaluation of the model, which were generated based on places with a long sampling effort where no records of the species were obtained. In this study, the potential distribution model of *B. henni* had a good performance (AUC = 0.90), with a 99% accuracy of all presence and absence records. The binary model under a threshold of 0.23 presented good sensitivity (positive accuracy rate: 0.95) and specificity (negative accuracy rate: 0.68). These results highlight the potential importance of these modelling applications in conservation actions on these freshwater systems.

IUCN currently defines the extent of occurrence as the area within the outermost limits of known or inferred occurrence for a species. Importantly, this area is not an estimate of the extent of occupied habitat or potential range of a taxon [59, 60]; instead, it measures the general geographic extension of the localities in which the species is found [61]. This method adopted by the IUCN measures the area of a convex polygon around the known species

records [60]. Nevertheless, this method is highly susceptible to sampling biases. For example, the polygon drawn around known locations may only represent a portion of a species geographic range, or it may also include a large area not inhabited by the species. Our results demonstrate that, for freshwater species in the Magdalena Basin, the proposed distribution areas from the ENM may be more representative than areas derived from the convex polygons (Fig 4). Furthermore, our approach to constraining the ENM predictions to hydrological basins also provides a strategy for identifying potentially suitable areas where the species may occur, but where it has not been detected yet. ENMs offer the opportunity to increase the objectivity and veracity of these evaluations by providing quantitative range estimates based on the relationship between species and their environment [62]. However, it is ideal that conservation assessments employ not only ENMs but also expert judgment [63, 64]; especially since the tropical Andes are suffering an additional loss in connectivity during the boom of dam building [65].

Many fish species require connectivity within the hydrological network to be able to visit feeding and spawning grounds in different sections of the fluvial network. This is why identifying freshwater species distribution at the level of the hydrological network is of utmost importance. In the present study, we used a Colombian endemic species as an example to show the importance of increasing the quality of the distribution hypotheses of freshwater species in order to generate inputs that contribute to the development of adequate management strategies for future conservation, protection, recovery and exploitation studies of freshwater fish species. Dam constructions in the Andean region are potentially isolating fish populations and the effect it will have on the species conservation within the basin is unknown [66].

Our results indicate that freshwater fish distributions inferred from ENMs can provide a realistic proxy about the geographic range of a species that can be used for several purposes including conservation actions. In the case of *B. henni*, its distribution is broader and with greater connectivity between basins than indicated by other current proposals such as the maps proposed by the IUCN. The inclusion of upstream variables in the most accurate models suggests that local (on site) and regional (upstream) information are both informative about the Sabaleta's niche. This situation might be a common property of species that engage in medium to long distance movements along the fluvial network. All this evidence shows the potential of ENMs to identify the environmental requirements of the species throughout the basin and to use the geographic projections of these models, incorporating expert's opinion, for making conservation decisions at the local level, as suggested by other studies [41, 42, 67]. We recommend future studies to include these geographic projections to inform spatial prioritization of areas for conservation that recognize connectivity as one of the key aspects of freshwater biodiversity.

## Supporting information

**S1 Fig. Graphic representation of the area occupied by a river within the basin and its percentage with respect to the basin's total area.** The shapefile of hydrographic basins and rivers of Colombia was obtained from IGAC (https://geoportal.igac.gov.co). All other products were produced by the authors and are copyright-free.
(TIF)

**S2 Fig. Correlation analysis among variables from the same category.** (A) Land use, (B) upstream climatic variables (average temperature and precipitation throughout the basin), (C) climatic data points and (D) set of variables that we recognized as most relevant for the species with no correlation (<0.75). For each predictor information, see S2 Table.
(TIF)

**S3 Fig. Cohen's Kappa quantification.** Kappa defines the precision of the prediction in relation to that expected by chance, it corresponds to the proportion of all test records that indicate agreement between the classifier and the observations.
(TIF)

**S4 Fig. Response curves of the most influential variables in the prediction of *B. henni* distribution in the Magdalena basin.** The first column represents the variables of the best model using both types of variables (on-site and upstream; model highlighted in bold in S3 Table) and in the second column, the response curves of the second best model using only on-site variables (see S3 Table). Response curves included in both models are presented side by side to facilitate comparison between them.
(PDF)

**S1 Table. Database with the records of presence and absence of *B. henni*, as well as the data used to train and validate the model.**
(PDF)

**S2 Table. List of analyzed predictors for the species.** The set of uncorrelated variables used to train the model are in bold type.
(PDF)

**S3 Table. Parametrization and performance of *Bycon henni* distribution models using different sets of aquatic predictor variables (on-site or on-site and upstream).** We evaluated the performance of models with different values of the regularization parameter from 1 to 4, and lineal (l), quadratic (q), product (p), threshold (t), and hinge (h) relationships. The number of free parameters ($k$); rescaled Akaike information criterion ($\Delta$AICc); Akaike weights (w. AIC) and the area under the curve (AUC), which is a measure of the model's performance in relation to a null model. Best model was chosen based on its AICc and its AUC value. In bold type the best overall model is highlighted and models are organized by the types of predictor variables used (on-site or on-site and upstream) and AICc (lowest to highest).
(PDF)

## Acknowledgments

This work was carried out through the agreement CT-2017-001714 between Empresas Públicas de Medellín and Universidad de Antioquia. In addition, the authors wish to thank Daniel Restrepo Santamaría, José David Botero Escalante and Merlin Hamp for their field work, to Andrés Felipe Galeano for his timely comments, Octavio Rojas-Soto for technical advice and especially Juliana Herrera Pérez for her contributions in the use of geoinformatic tools, field work and discussions that allowed an improved development of this proposal. The final version of this manuscript was improved also by useful comments from the Editor and one reviewer.

## Author Contributions

**Conceptualization:** Luz Jiménez-Segura, Carlos A. Rogéliz, Juan L. Parra.

**Data curation:** Daniel Valencia-Rodríguez.

**Formal analysis:** Daniel Valencia-Rodríguez.

**Investigation:** Juan L. Parra.

**Methodology:** Daniel Valencia-Rodríguez, Juan L. Parra.

**Project administration:** Luz Jiménez-Segura.

**Resources:** Juan L. Parra.

**Supervision:** Luz Jiménez-Segura, Juan L. Parra.

**Writing – original draft:** Carlos A. Rogéliz.

**Writing – review & editing:** Daniel Valencia-Rodríguez, Luz Jiménez-Segura, Juan L. Parra.

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
