## [Decision Letter · Decision Letter 0]

8 Oct 2020

PONE-D-20-23545

Ecological niche modeling as an effective tool to predict the distribution of freshwater organisms: the case of the Sabaleta (CHARACIDAE: *Brycon henni*)

PLOS ONE

Dear Dr. Valencia-Rodríguez,

Thank you for submitting your manuscript to PLOS ONE. After careful consideration, we feel that it has merit but does not fully meet PLOS ONE’s publication criteria as it currently stands. Therefore, we invite you to submit a revised version of the manuscript that addresses the points raised during the review process.

I would firstly like to apologise for the delay in issuing this decision. We were waiting for one important reviewer who agreed but finally did not deliver. One expert in the field and myself have reviewed the manuscript. Although we find the study interesting, there is a number of questions and concerns with it, so I am recommending that you undertake a major revision of your manuscript.

I invite you to carefully respond to the editor and reviewers' comments and revise your manuscript accordingly. Your manuscript will be sent for a second round of revision, and it is therefore imperative you provide thorough responses/revisions to each of the comments and suggestions below.

Editor’s Comments

The manuscript represents an interesting and valuable study exemplifying the application of SDMs to freshwater species. Positive aspects of the manuscript include (i) the inclusion of upstream variables along with on-site predictors, and (ii) a good modelling exercise, including the identification of an optimal regularization parameter and feature types, as well as the use of independent occurrence data and absence records to evaluate the performance of the model. Additionally, I have to stress the value of such biogeographical studies conducted in neotropical systems, which are underrepresented in the literature. Said that, although I found the subject very interesting there are some issues that need to be addressed by the authors.

First, one of the main conclusions of the study is the utility of ENMs in conservation and decision-making at the national level. However, a number of papers have previously shown the utility of niche models in a variety of conservation assessments to address pressing conservation problems (see e.g. Guisan et al. 2013, Ecol. Letters 16: 1424-1435). Hence, authors should adjust the conclusions of the study to the state of the art in ENM and conservation.

Second, another important conclusion of the study is the improvement of ENMs that incorporate upstream variables to model freshwater species. However, one might wonder to what extent it is possible to obtain models as good as the obtained here but just with on-site predictors (which are more easily obtained, as they are already available for every cell across the globe). Thus, and in agreement with the anonymous reviewer, I think that it would be interesting to compare the predictive ability and model support among species distribution models based on the two different sets of predictors (on-site variables vs. on-site plus upstream variables). See e.g. an example in Abellán et al. 2012, J. Biogeogr. 39: 970-983 comparing the relative performance of models based on two sets of predictors.

Other minor comments:

- L40: Review sentence

- L53-54: Cite the references as numbers in square brackets.

- L136: The origin of climate data (i.e. Worldclim) should be specified.

- L260-261: The explanation of bold/regular style in Table 1 should be included in the Table title. Additionally, the most relevant predictors were 5 and not 3, according to Table 1.

- L263: As authors explain in L356-368, the IUCN’s range map was obtained from a convex polygon around known species records and not from a modelling approach. Then, please, the use of “model” should be avoid throughout the text when referring to the IUCN distribution map (see also e.g. L209, 263 and 384).

- L598: Are these the most relevant predictors or just the set of the uncorrelated variables used to train the model?

- While the Figure 1 is a good illustration of the differences between using catchments and the river network approaches, I think it is not relevant in this study to be included in the main text. I suggest moving it to Supporting Information.

We look forward to receiving your revised manuscript.

Kind regards,

Pedro Abellán

Academic Editor

PLOS ONE

Journal Requirements:

2. We note that Figure 1, 2, 4, 5 in your submission contain map images which may be copyrighted. All PLOS content is published under the Creative Commons Attribution License (CC BY 4.0), which means that the manuscript, images, and Supporting Information files will be freely available online, and any third party is permitted to access, download, copy, distribute, and use these materials in any way, even commercially, with proper attribution. For these reasons, we cannot publish previously copyrighted maps or satellite images created using proprietary data, such as Google software (Google Maps, Street View, and Earth). For more information, see our copyright guidelines: http://journals.plos.org/plosone/s/licenses-and-copyright.

2.1.    You may seek permission from the original copyright holder of Figure 1, 2, 4, 5 to publish the content specifically under the CC BY 4.0 license. 

2.2.    If you are unable to obtain permission from the original copyright holder to publish these figures under the CC BY 4.0 license or if the copyright holder’s requirements are incompatible with the CC BY 4.0 license, please either i) remove the figure or ii) supply a replacement figure that complies with the CC BY 4.0 license. Please check copyright information on all replacement figures and update the figure caption with source information. If applicable, please specify in the figure caption text when a figure is similar but not identical to the original image and is therefore for illustrative purposes only.

Reviewers' comments:

Reviewer's Responses to Questions

**Comments to the Author**

1. Is the manuscript technically sound, and do the data support the conclusions?

Reviewer #1: Yes

2. Has the statistical analysis been performed appropriately and rigorously? 

Reviewer #1: Yes

3. Have the authors made all data underlying the findings in their manuscript fully available?

Reviewer #1: Yes

4. Is the manuscript presented in an intelligible fashion and written in standard English?

Reviewer #1: Yes

5. Review Comments to the Author

Reviewer #1: This is an interesting case study trying to provide accurate potential distributions of an endemic fish of the Colombian Andes (Brycon henni), using fine scale environmental variables. From my particular point of view, it is not very novel (a number of papers have been conducted with similar purposes and variables), but it is relatively well-written and correct from a methodological perspective. I particularly like the way in which the model was validated (using new sampling and the absences recorded in the field).

It is not surprising that the use of fine scale (on-site, 1x1 km) environmental predictors provide accurate predictions. The question remain is how far the use of upstream conditions accumulated for the entire basin obtained from data at this same resolution could improve these models. Thus, I would suggest the authors to consider this point as this could add more interest to their paper. For this, once a first model was carried out using only on-site variables, it should be interesting to include variables summarizing the upstream conditions and then, to evaluate the relative performance of these two models.

From a methodological perspective it must be clear that this species is distributed exclusively in the Magdalena basin. In the same way I was wondering if there are records in the east part of the Magdalena river or the south of the study area (Bucaramanga, Pasto, etc.. that are included in the map from the IUCN) that were not considered for this model. This is important as if this species is occurring in these areas, some environmental suitable conditions could be missing in the model, providing misleading results (see Sánchez-Fernández et al., 2011; Diversity & Distributions, 17: 163-171).

The importance of specific variables to determine occurrence probability must be interpreted carefully, as these variables were picked up from a pool of correlated variables.

Lastly, please, review carefully the format of the reference list.

Minor points:

L83-83. Please, include that this is not an endangered species as it is considered at Least Concern (LC) by the IUCN.

L 99. Figure 1 is not necessary at all. I would send it to supplementary material.

L 131. Please, could you specify these requirements?

L172. Why only the records presents in GBIF?

L212. Please, provide details on how the IUCN map is constructed. This is relevant to interpret the comparison.

L 258. Table 1. Percentage contribution (PC) and permutation importance (PI) must be explained, at least, in the methods.

L265. Please, specify what occurrence records were used to verify the number of occurrence records that were correctly predicted by each proposal. Were used all records or only the 20% reserved to validate the model? What about absences?

L209. There are not other models to compare with, simply the map from the IUCN.

6. PLOS authors have the option to publish the peer review history of their article (what does this mean?). If published, this will include your full peer review and any attached files.

Reviewer #1: No

---

## [Author Response · Author response to Decision Letter 0]

2 Dec 2020

Dear Editor,

Hope you are doing well. We would like to thank the reviewers and you for the meticulous evaluation of our manuscript. We are encouraged by your constructive criticism and positive feedback. We hope our manuscript makes an important contribution to PLoS One. We have addressed and answered in detail all the comments and are convinced this new version represents an improvement that incorporates a new analysis. We hope these changes meet the editor and reviewers’ expectations. 

The most important changes are related to providing more detail and clarification of methodological details (e.g., relative importance of variables, ), a new analyses suggested by reviewer 1 that included only in situ variables in order to verify if we could achieve the same performance using only these variables, and finally, an improved version of the discussion. 

Response to each comment is below in blue.

Editor’s Comments

Comment: The manuscript represents an interesting and valuable study exemplifying the application of SDMs to freshwater species. Positive aspects of the manuscript include (i) the inclusion of upstream variables along with on-site predictors, and (ii) a good modelling exercise, including the identification of an optimal regularization parameter and feature types, as well as the use of independent occurrence data and absence records to evaluate the performance of the model. Additionally, I have to stress the value of such biogeographical studies conducted in neotropical systems, which are underrepresented in the literature. Said that, although I found the subject very interesting there are some issues that need to be addressed by the authors.

First, one of the main conclusions of the study is the utility of ENMs in conservation and decision-making at the national level. However, a number of papers have previously shown the utility of niche models in a variety of conservation assessments to address pressing conservation problems (see e.g. Guisan et al. 2013, Ecol. Letters 16: 1424-1435). Hence, authors should adjust the conclusions of the study to the state of the art in ENM and conservation.

Response: We agree that the utility of niche models in conservation is not our main contribution and that it has been widely acknowledge in previous studies. Nonetheless, our results support this general conclusion and provide particular guidelines to consider for making conservation decisions, such as acknowledging the key role of connectivity for freshwater species. We modified the paragraph to better represent our contributions and acknowledge previous ones (see last paragraph of discussion).

Second, another important conclusion of the study is the improvement of ENMs that incorporate upstream variables to model freshwater species. However, one might wonder to what extent it is possible to obtain models as good as the obtained here but just with on-site predictors (which are more easily obtained, as they are already available for every cell across the globe). Thus, and in agreement with the anonymous reviewer, I think that it would be interesting to compare the predictive ability and model support among species distribution models based on the two different sets of predictors (on-site variables vs. on-site plus upstream variables). See e.g. an example in Abellán et al. 2012, J. Biogeogr. 39: 970-983 comparing the relative performance of models based on two sets of predictors.

Response: Thank you for your positive comments and constructive criticism. We include now a new analysis where we generate species distribution models using only on-site variables in order to verify if these models can have the same or improved predictive accuracy than models using both on-site and upstream variables. We conclude, as we had envisioned, that upstream variables provide complementary information which adds not only predictive accuracy but likely biological information (e.g., response curves) that is valuable to better understand the ecology of these freshwater fishes. We also made adjustments to the methods section "relative importance of environmental predictors (Lines 195-212)" and we aggregated supplementary information (S3_Table and S4_Fig) including the results of our new analyses. 

Other minor comments:

L40: Review sentence

Response: Sentence was changed to: “Both on-site (annual precipitation, minimum temperature of coldest month) and upstream variables (open waters, average minimum temperature of the coldest month and average precipitation seasonality) were included in the models with the highest predictive accuracy”.

L54: Cite the references as numbers in square brackets.

Response: Done.

L147: The origin of climate data (i.e. Worldclim) should be specified.

Response: Data source is now specified.

L292-293: The explanation of bold/regular style in Table 1 should be included in the Table title. Additionally, the most relevant predictors were 5 and not 3, according to Table 1.

Response: Thank you, this has been corrected.

L296: As authors explain in L356-368, the IUCN’s range map was obtained from a convex polygon around known species records and not from a modelling approach. Then, please, the use of “model” should be avoid throughout the text when referring to the IUCN distribution map (see also e.g. L305, 338 and 427).

Response: We replaced the word model by map when referring to the IUCN map.

L668-669: Are these the most relevant predictors or just the set of the uncorrelated variables used to train the model?

Response: We changed the sentence to “The set of uncorrelated variables used to train the model are in bold type”.

While the Figure 1 is a good illustration of the differences between using catchments and the river network approaches, I think it is not relevant in this study to be included in the main text. I suggest moving it to Supporting Information.

Response: We sent Figure 1 to supplementary material (S1 Fig).

Reviewer #1: This is an interesting case study trying to provide accurate potential distributions of an endemic fish of the Colombian Andes (Brycon henni), using fine scale environmental variables. From my particular point of view, it is not very novel (a number of papers have been conducted with similar purposes and variables), but it is relatively well-written and correct from a methodological perspective. I particularly like the way in which the model was validated (using new sampling and the absences recorded in the field).

It is not surprising that the use of fine scale (on-site, 1x1 km) environmental predictors provide accurate predictions. The question remain is how far the use of upstream conditions accumulated for the entire basin obtained from data at this same resolution could improve these models. Thus, I would suggest the authors to consider this point as this could add more interest to their paper. For this, once a first model was carried out using only on-site variables, it should be interesting to include variables summarizing the upstream conditions and then, to evaluate the relative performance of these two models.

From a methodological perspective it must be clear that this species is distributed exclusively in the Magdalena basin. In the same way I was wondering if there are records in the east part of the Magdalena river or the south of the study area (Bucaramanga, Pasto, etc.. that are included in the map from the IUCN) that were not considered for this model. This is important as if this species is occurring in these areas, some environmental suitable conditions could be missing in the model, providing misleading results (see Sánchez-Fernández et al., 2011; Diversity & Distributions, 17: 163-171).

Response: When we verify the occurrences data, we identified that records reported in areas such as Santander, Nariño, Choco etc ... presented inconsistencies (e.g., municipality or department) did not coincide with the assigned coordinates. In other cases, we verify the lots deposited in the biological collections and the sample was not found or the species present taxonomic identification problems.

The importance of specific variables to determine occurrence probability must be interpreted carefully, as these variables were picked up from a pool of correlated variables.

Lastly, please, review carefully the format of the reference list.

Minor points:

L86-87. Please, include that this is not an endangered species as it is considered at Least Concern (LC) by the IUCN.

Response: We specify now in the Introduction: “This species is considered of minor concern (LC) by the International Union for Conservation of Nature (IUCN)”.

L 99. Figure 1 is not necessary at all. I would send it to supplementary material.

Response: We sent Figure 1 to supplementary material (S1 Fig).

L 136-140. Please, could you specify these requirements?

Response: The sentence was updated to: “The final database contained 186 records after discarding duplicate records (all records within the same 30 second resolution pixel), dubious records from outside the Magdalena River Basin or whose description (e.g., municipality or department) did not coincide with the assigned coordinates (435 records discarded in total)”.

L175. Why only the records presents in GBIF?

Response: As far as we are concerned, all locality data present in biological collections is available through GBIF. Accessibility to the data is easier through GBIF rather than accessing each project individually. All the information uploaded to the Colombian Biodiversity Information System (SIB) is immediately linked to the GBIF.

L232-235. Please, provide details on how the IUCN map is constructed. This is relevant to interpret the comparison.

Response: We included this sentence in the methods section: “The mapping protocol used by the IUCN consists of collecting point locality data for each species, identifying all sub-basins at resolution 8 following the HydroBASINS database [38], and optionally, other sub-basins can be included where the species native presence is potential based on literature or expert knowledge [39]”

L 258. Table 1. Percentage contribution (PC) and permutation importance (PI) must be explained, at least, in the methods.

Response: The explanation is included in the section “Relative importance of environmental predictors”. Most of this section was rephrased in order to better explain these two metrics and the new analyses included. 

L303-306. Please, specify what occurrence records were used to verify the number of occurrence records that were correctly predicted by each proposal. Were used all records or only the 20% reserved to validate the model? What about absences?

Response: We used 30 occurrence records for evaluation. We now specify this explicitly in the text: “However, when we verified the number of occurrence records used to evaluate the model (total of 30 occurrences) that were correctly predicted by each proposal, we obtained that the IUCN map was correct in 39% while our model was correct in 99% of the cases”.

L238-239. There are not other models to compare with, simply the map from the IUCN. 

Response: To our knowledge, there are no other published proposed distributions for this species, except coarse generalizations (e.g., Colombia, or South America: Trans-Andean river basins of Colombia).

---

## [Decision Letter · Decision Letter 1]

29 Dec 2020

PONE-D-20-23545R1

Ecological niche modeling as an effective tool to predict the distribution of freshwater organisms: the case of the Sabaleta *Brycon henni* (Eigenmann, 1913)

PLOS ONE

Dear Dr. Valencia-Rodríguez,

Thank you for submitting your manuscript to PLOS ONE. After careful consideration, we feel that it has merit but does not fully meet PLOS ONE’s publication criteria as it currently stands. Therefore, we invite you to submit a revised version of the manuscript that addresses the points raised during the review process.

The previous reviewer and myself have reviewed the new version of the manuscript, and we both agree that the paper is much improved, and that the authors have addressed the previous comments seriously and effectively. The reviewer has recommended publication, but also suggests some minor yet important revisions to your manuscript that should be addressed. Therefore, I invite you to respond to the reviewer' comments and revise your manuscript. Overall, the reviewer has pointed the need of discuss the relative importance of the variables, and the potential confounding use of the word “connectivity” when referring to upstream or watershed factors (note that stream connectivity usually refers to the longitudinal connections or pathways along the length of a stream).

We look forward to receiving your revised manuscript.

Kind regards,

Pedro Abellán

Academic Editor

PLOS ONE

Reviewers' comments:

Reviewer's Responses to Questions

**Comments to the Author**

1. If the authors have adequately addressed your comments raised in a previous round of review and you feel that this manuscript is now acceptable for publication, you may indicate that here to bypass the “Comments to the Author” section, enter your conflict of interest statement in the “Confidential to Editor” section, and submit your "Accept" recommendation.

Reviewer #1: (No Response)

2. Is the manuscript technically sound, and do the data support the conclusions?

Reviewer #1: Yes

3. Has the statistical analysis been performed appropriately and rigorously? 

Reviewer #1: Yes

4. Have the authors made all data underlying the findings in their manuscript fully available?

Reviewer #1: Yes

5. Is the manuscript presented in an intelligible fashion and written in standard English?

Reviewer #1: Yes

6. Review Comments to the Author

Reviewer #1: The paper has been improved considerably. However, I would like to see some text on the "relative" importance of specific variables on the distribution of the target species (as they come from a pool of higly correlated variables). In the same way, I also think that the concept of "conectivity" is mixed in the last paragraph of the discusion. The authors are mixing the imporance of conectivity among sites for the conservation of freshwater fauna and the importance of variables that operate at basin scales to predict the species occurence.

7. PLOS authors have the option to publish the peer review history of their article (what does this mean?). If published, this will include your full peer review and any attached files.

Reviewer #1: No

---

## [Author Response · Author response to Decision Letter 1]

12 Feb 2021

PONE-D-20-23545R1

Ecological niche modeling as an effective tool to predict the distribution of freshwater organisms: the case of the Sabaleta Brycon henni (Eigenmann, 1913)

February 12, 2021

Pedro Abellán

Academic Editor

PLOS ONE

Dear Editor,

On behalf of my coauthors, I would like to thank you and the reviewers for the constructive criticism. We have addressed all comments and hope this version of the manuscript meets your expectations.

Response to each comment is below in blue.

Editor’s Comments

The previous reviewer and myself have reviewed the new version of the manuscript, and we both agree that the paper is much improved, and that the authors have addressed the previous comments seriously and effectively. The reviewer has recommended publication, but also suggests some minor yet important revisions to your manuscript that should be addressed. Therefore, I invite you to respond to the reviewer' comments and revise your manuscript. Overall, the reviewer has pointed the need of discuss the relative importance of the variables, and the potential confounding use of the word “connectivity” when referring to upstream or watershed factors (note that stream connectivity usually refers to the longitudinal connections or pathways along the length of a stream).

Response: The second and third paragraph of the discussion, in addition to Table 1, include all information about the relative importance of variables, and whether they can be interpreted as complementary or redundant. The most important variables in the model included annual precipitacion, precipitation seasonality and minimum temperature – both on site and upstream -, and percent open water. We included a sentence in the third paragraph that describes how these variables relate to relative species’ occurrence rate: “Response curves of these variables indicate that the Sabaleta can be found in sites with relatively high annual precipitation (> 2500 mm), little percent open water (<10 %), profiles of minimum temperatures that include both low and high elevation sites (~10-20 C), and relatively little precipitation seasonality (S4 Fig).” 

We also edited the use of the word "Connectivity" in the last paragraph of the discussion. Our intention was to emphasize the importance of including variables at multiple scales (on site and upstream) in ENM of freshwater species. 

Comments to the Author

Reviewer #1: The paper has been improved considerably. However, I would like to see some text on the "relative" importance of specific variables on the distribution of the target species (as they come from a pool of higly correlated variables). In the same way, I also think that the concept of "conectivity" is mixed in the last paragraph of the discusion. The authors are mixing the imporance of conectivity among sites for the conservation of freshwater fauna and the importance of variables that operate at basin scales to predict the species occurence.

Response: Thank you, we rephrased these sentences to avoid confusion.

---

## [Editor Report · Decision Letter 2]

16 Feb 2021

Ecological niche modeling as an effective tool to predict the distribution of freshwater organisms: the case of the Sabaleta *Brycon henni* (Eigenmann, 1913)

PONE-D-20-23545R2

Dear Dr. Valencia-Rodríguez,

We’re pleased to inform you that your manuscript has been judged scientifically suitable for publication and will be formally accepted for publication once it meets all outstanding technical requirements.

Kind regards,

Pedro Abellán

Academic Editor

PLOS ONE
---

## [Editor Report · Acceptance letter]

22 Feb 2021

PONE-D-20-23545R2 

Ecological niche modeling as an effective tool to predict the distribution of freshwater organisms: the case of the Sabaleta *Brycon henni* (Eigenmann, 1913) 

Dear Dr. Valencia-Rodríguez:

I'm pleased to inform you that your manuscript has been deemed suitable for publication in PLOS ONE. Congratulations! Your manuscript is now with our production department. 

Kind regards, 

on behalf of

Dr. Pedro Abellán 

Academic Editor

PLOS ONE